

# A simple spectrophotometric evaluation method for the hydrophobic anticancer drug paclitaxel

Ken Sugo[1] and Mitsuhiro Ebara[1,2,3]

[1] Graduate School of Pure and Applied Sciences, University of Tsukuba, Tsukuba, Ibaraki, Japan
[2] International Center for Materials Nanoarchitectonics, National Institute for Materials Science, Tsukuba, Ibaraki, Japan
[3] Graduate School of Industrial Science and Technology, Tokyo University of Science, Katsushika, Tokyo, Japan

Corresponding author
Mitsuhiro Ebara,
EBARA.Mitsuhiro@nims.go.jp

## ABSTRACT

In this work, we demonstrate a simple spectrophotometry approach to more accurately quantify and measure paclitaxel (PTX) concentrations. PTX cannot be precisely quantified when mixed with an aqueous solvent, and carries the risk of undergoing crystal precipitation. It is likely that PTX undergoes numerous interactions with aqueous solvents and enters a supersaturated state due to its low solubility. Therefore, a quantitative method is required to measure PTX for quality control before clinical use. Although several high-performance liquid chromatography (HPLC) methods have been reported to date, not all medical facilities have a clinical laboratory with such HPLC devices and analysis techniques. Spectroscopy is a simple and convenient method; however, calibration standards are prepared with an organic solvent, such as methanol and acetonitrile, which, when mixed with PTX, can cause solvent effects that lead to inaccurate results.
We generated a calibration curve of PTX at various concentrations (40%, 50%, 60%, 70%, 80%, 90% and 100%) of methanol and evaluated the relative error from HPLC results. The optimum methanol concentration for quantification of PTX was 65.8%, which corresponded to the minimum relative error. The detection limit and quantification limit were 0.030 μg/mL and 0.092 μg/mL, respectively. It was possible to predict the PTX concentration even when polyoxyethylene castor oil and anhydrous ethanol were added, as in the commercially available PTX formulation, by diluting 32-fold with saline after mixing. Our findings show that PTX can be more accurately quantified using a calibration curve when prepared in a methanol/water mixture without the need for special devices or techniques.

# INTRODUCTION

To properly use an anticancer drug in clinical settings, it is necessary to periodically verify its stability after mixing (*Badea et al., 2004*; *Kawashima et al., 2015*). The hydrophobic anticancer drug paclitaxel (PTX), isolated from the bark of *Taxus brevifolia*, is one of the most important anticancer drugs and is effective against a variety of human cancers, including breast and ovarian cancer (*Huizing et al., 1995*; *Wani et al., 1971*). It is extremely

lipophilic (log $P$ = 3.5) and practically insoluble in water (0.3 ± 0.02 µg/mL) (*Ahmad et al., 2013*), and is therefore commercially available as a suspension in polyoxyethylene castor oil and anhydrous ethanol. Inconveniently, PTX cannot be precisely quantified when mixed with an aqueous solvent such as saline or glucose injection before use, and carries the risk of undergoing crystal precipitation (*Kawashima et al., 2015*). It is likely that PTX undergoes numerous interactions with aqueous solvents and enters a supersaturated state due to its low solubility, although the details are unclear (*Finney et al., 1980*; *Ohno, Abe & Tsuchida, 1978*). Therefore, a quantitative method is required to measure PTX for quality control.

Although several reversed-phase high-performance liquid chromatography (HPLC) methods for quantification have been reported to date (*Bonde, Bonde & Prabhakar, 2019*; *Choudhury et al., 2014*; *Khan et al., 2016*; *Kim et al., 2005*; *Wang et al., 2003*; *Xavier Junior et al., 2016*; *Yonemoto et al., 2007*), not all medical facilities have a clinical laboratory with such HPLC devices and analysis techniques. In addition, PTX must be used immediately after mixing, and therefore requires a simple, fast, and precise method that can be conducted in the hospital or clinic rather than being outsourced.

Spectroscopy is a simple and convenient method (*Heydari et al., 2016*). However, there are important reasons why HPLC is recommended for measuring PTX. Although spectroscopic drug release evaluation can be successfully undertaken if the test specimen can be prepared with a solvent using the same conditions as those used to generate the standard calibration curve (*Kesarwani, Tekade & Jain, 2011*), our study focuses on the clinical quality control of non-single component PTX formulations. Namely, while the PTX formulation is diluted with an aqueous solvent in clinical use, as described above, calibration standards for spectroscopy must be accurately prepared with an organic solvent such as methanol and acetonitrile, which, if mixed with PTX, would cause solvent effects and lead to inaccurate results. The HPLC method does not have this problem because the solvent is displaced by the HPLC mobile phase. Conventionally, the test specimen is resuspended in an appropriate solvent after lyophilization and diluted with methanol, dimethyl sulfoxide (DMSO), or $N,N$-dimethylformamide (DMF) to prevent interactions with the solvent from affecting the analysis (*Ni et al., 2001*). However, the lyophilization process causes both freezing and drying stresses, which can cause deactivation and degradation of the drug (*Wang, 2000*). Furthermore, direct dilution of a test specimen with an organic solvent may also cause unknown transformation of the sample.

Here, we demonstrate that PTX mixed with aqueous solvents can be quantified spectrophotometrically using a calibration curve when prepared in a methanol/water mixture without the need for special devices or techniques (Fig. 1).

## MATERIALS AND METHODS

### Reagents

Paclitaxel powder and injectable formulation were purchased from Tokyo Chemical Industry (Tokyo, Japan) and Bristol-Myers Squibb Company (Tokyo, Japan), respectively. Methanol, acetonitrile, anhydrous ethanol, and polyoxyethylene (10) castor oil were all

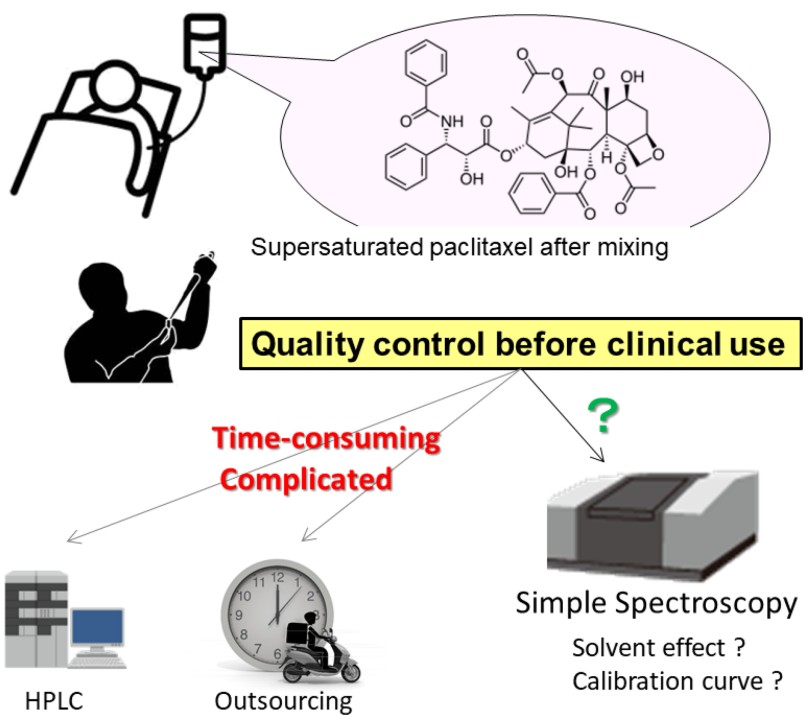

Figure 1 Schematic illustration of the background (Graphics credit: Silhouette AC, Wikipedia, PictArts and Kenkyu.net).               

purchased from Fujifilm Wako Pure Chemical (Osaka, Japan). Normal saline was purchased from Otsuka Pharmaceutical Factory (Tokushima, Japan).

## Calibration curve of PTX prepared using various concentrations of methanol

Each concentration of PTX (0.313, 0.625, 1.25 and 2.50 μg/mL) was prepared with 40%, 50%, 60%, 70%, 80%, 90% and 100% methanol. Absorbance was measured using an ultraviolet (UV) and visible spectrophotometer (UV-1800; Shimadzu, Kyoto, Japan) at a wavelength of 230 nm after blank correction with each solvent. Each absorbance reading was plotted against the corresponding known PTX concentration to generate a calibration curve. This experiment was replicated three times at room temperature.

## Preparation of saturated PTX in saline

Approximately 2.5 mg of PTX was added to 10 mL of saline, vortexed for 30 s and mixed by rotating for 24 h at 30 rpm at 4 °C. The supernatant was collected after centrifuging at 9,000×$g$ for 10 min at 4 °C.

## Quantification by HPLC

A series of standard solutions of different known concentrations of PTX (0.313, 0.625, 1.25 and 2.50 μg/mL) in methanol and the PTX-saturated saline solution, prepared above, were injected (20 μL each) into an octadecylsilyl silica gel column (5 μm, φ4.6 mm × h 250 mm, Osaka Soda, Osaka, Japan) of an Elite LaChrom HPLC system

(Hitachi High-Technologies, Tokyo, Japan) with 50% acetonitrile aqueous solution as the mobile phase and a flow rate of 1.2 mL/min (*Yonemoto et al., 2007*). The eluent was monitored at 230 nm. The peak area of PTX in each standard solution was measured and plotted against the PTX concentration to generate a calibration curve. The concentration of PTX in each test specimen (the PTX-saturated saline solution) was subsequently determined using the same conditions as those used to generate the standard calibration curve.

## Quantification by spectrophotometry

The concentration of PTX-saturated saline was also measured spectrophotometrically at a wavelength of 230 nm using a calibration curve prepared based on the methanol dilution series (40%, 50%, 60%, 70% and 80%). This experiment was replicated three times at room temperature. The quantitative concentrations were compared to those obtained using HPLC. Accuracy ($R$) indicates the relative error, which was defined as the deviation from the HPLC results, and was calculated as follows:

$$R = (Cs - Cr)/Cr \tag{1}$$

where $Cs$ is the quantitative concentration measured spectrophotometrically and $Cr$ is the concentration measured using HPLC.

## Practical simulation

For quality control against the commercially available PTX formulation, the method was examined in the presence of polyoxyethylene castor oil and anhydrous ethanol. First, 30 mg of PTX was added to 2.5 mL of polyoxyethylene castor oil and 2.5 mL of anhydrous ethanol. The mixture was diluted 100-fold with saline, as in clinical use, and then further diluted from 2-fold to 1,024-fold (final: 200–102,400-fold dilution) with saline to prepare a 2-fold dilution series. The absorbance of each diluted solution was measured spectrophotometrically at a wavelength of 230 nm. The PTX concentration in each diluted solution was determined using the HPLC method described above. Reference solutions were prepared in the same manner with the exclusion of PTX, and the absorbance was measured spectrophotometrically at a wavelength of 230 nm.

## Verification of method accuracy

Injectable PTX formulation was quantitatively analyzed using the method established in this study, the accuracy of which was verified. The PTX formulation was diluted 100-fold with saline, as in clinical use, and then further diluted 32-fold (final: 3,200-fold dilution) with saline. This further 32-fold dilution is unique to this study and showed reasonable values in spectroscopy. The PTX concentration was measured spectrophotometrically in the same manner as that described above and compared to the results obtained using HPLC. The experiment was replicated five times and the unpaired *t*-test was performed using Excel 2010 (Microsoft, Redmond, WA, USA). Differences between the spectrometry and HPLC values were considered statistically significant when the *p*-value was less than 0.05.
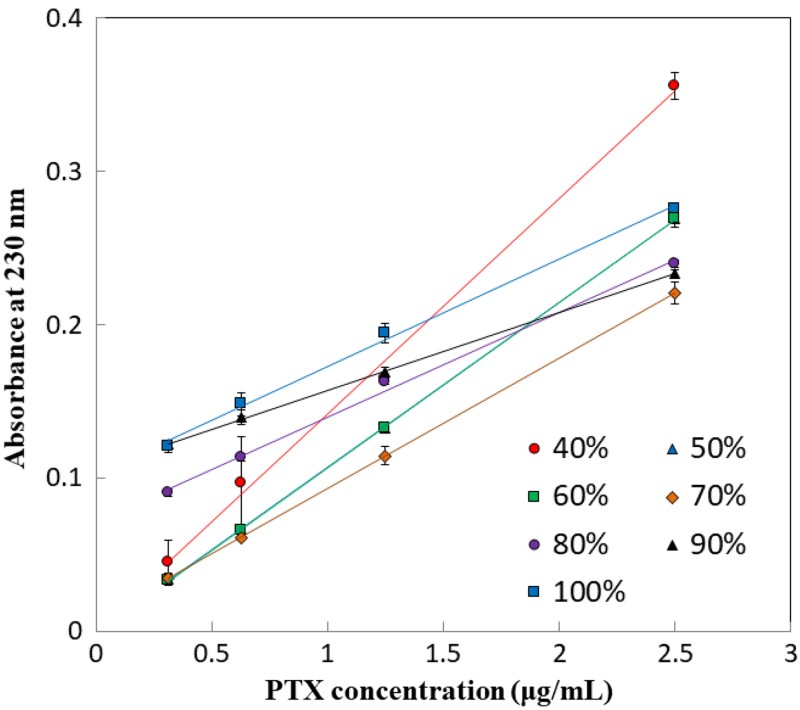

**Figure 2 Calibration curves of PTX.**

## RESULTS

### Calibration curve

We found that parameters of the PTX calibration curve, including slope and intercept, varied depending on the solvent used (Fig. 2; Table 1). The calibration curves showed high absorbance values at low concentrations of less than 1 µg/mL when the methanol concentration was 80% or higher. While the values obtained at 50% and 60% methanol were comparable, a marked change in the calibration curve was observed at 70–80% methanol. The findings suggest that quantification results for PTX, particularly at low concentrations, may vary substantially if experiments are conducted in solvents that differ to those used for calibration.

### Optimum methanol concentration

According to HPLC, the average concentration of triplicate tests in the PTX-saturated saline solution was 0.731 ± 0.0438 µg/mL. The concentration in the same PTX-saturated saline solution was also measured spectrophotometrically using each calibration curve, as shown in Fig. 2. The absorbance of the PTX-saturated saline solution at a wavelength of 230 nm was 0.058–0.074, which could not be measured using calibration curves prepared based on dilution in 90% or 100% methanol. Table 2 shows the spectroscopic quantitative concentrations calculated using each calibration curve, including the accuracy (*R*). An *R* value close to "0" indicates that the results from the two methods are comparable.

**Table 1 Details of the calibration curves.**

| Methanol concentration (%) | Slope | Intercept | Correlation coefficient |
|---|---|---|---|
| 40 | 0.1405 | 0.0010 | 0.9963 |
| 50 | 0.1079 | −0.0010 | 0.9999 |
| 60 | 0.1080 | −0.0014 | 0.9999 |
| 70 | 0.0851 | 0.0079 | 1.0000 |
| 80 | 0.0681 | 0.0717 | 0.9967 |
| 90 | 0.0508 | 0.1066 | 0.9998 |
| 100 | 0.0702 | 0.1024 | 0.9968 |

**Table 2 Spectroscopic quantitative concentration of PTX.**

| Methanol concentration (%) | Quantitative concentration (μg/mL) | Accuracy (relative error) |
|---|---|---|
| 40 | 0.371 ± 0.176 | −0.492 ± 0.241 |
| 50 | 0.598 ± 0.00751 | −0.181 ± 0.0103 |
| 60 | 0.600 ± 0.0115 | −0.179 ± 0.0157 |
| 70 | 0.653 ± 0.0384 | −0.106 ± 0.0525 |
| 80 | 0.0335 ± 0.00354 | −0.954 ± 0.00484 |

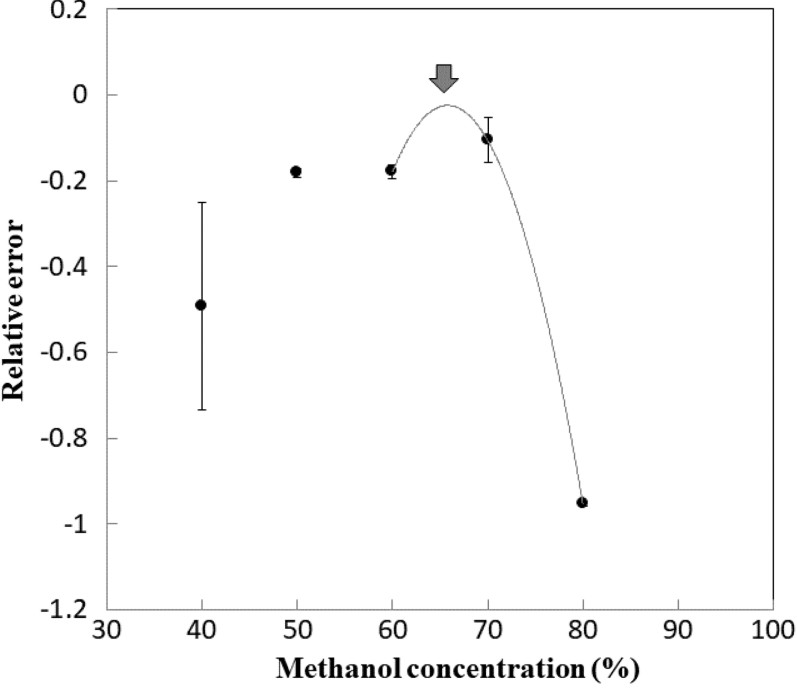

**Figure 3 Relative error of quantitative PTX.**     

Figure 3 shows the correlation between the methanol concentration and relative error from the HPLC results. The x-intercept of the approximate curves indicates the concentration of methanol at which PTX concentrations in saline were comparable to those obtained using HPLC. However, because the x-intercept could not be determined in
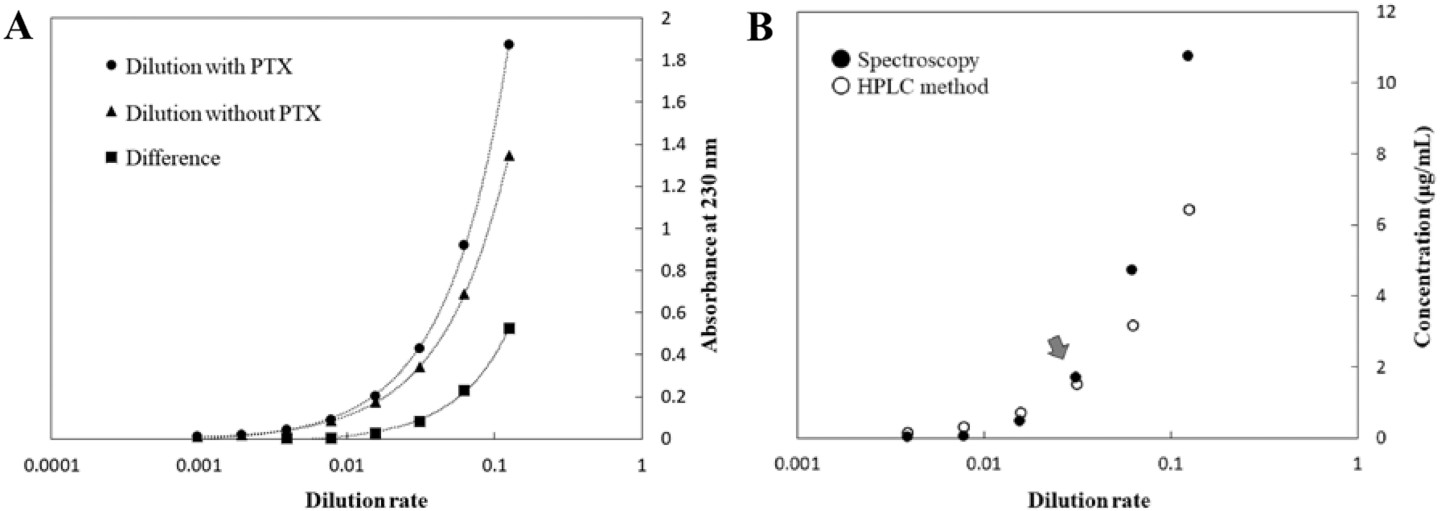

**Figure 4 Absorbance of each dilution and PTX concentration.** (A) Absorbance of each dilution solution with or without PTX and the difference in absorbance, and (B) PTX concentration in each dilution solution compared to that determined using HPLC, which was calculated using the differential absorbance obtained in (A) and a calibration curve prepared based on dilution in 65.8% methanol.

this study, the solution to the approximate curve equation, 65.8%, corresponding to the minimum relative error (−0.0174; Fig. 3, arrow), was identified as the optimum methanol concentration for quantification of PTX. The calibration curve prepared using 0.313, 0.625, 1.25 and 2.50 µg/mL PTX in 65.8% methanol as the solvent can therefore be expressed using the regression curve ($r^2$ = 0.9998) with the slope 0.0486 and the intercept 0.0032 (Fig. S1). The detection limit and quantification limit of PTX were 0.030 µg/mL and 0.092 µg/mL, respectively.

## Simulation

Paclitaxel formulations contain polyoxyethylene castor oil and anhydrous ethanol, and the effects of these solvents should be considered when quantifying PTX. The concentrations of PTX, polyoxyethylene castor oil, and anhydrous ethanol used in this experiment were the same as those in the commercially available formulation (*Chen et al., 2001*). Figure 4A shows the absorbance at each dilution with or without PTX and the difference in absorbance, which would indicate the absorbance derived from PTX. Figure 4B shows the PTX concentration in each dilution compared to that obtained using HPLC, which was calculated using the differential absorbance obtained in Fig. 4A and a calibration curve prepared using 65.8% methanol, the optimal concentration for quantification of PTX. There was higher correlation between the results obtained using the calibration curve and HPLC at a 32-fold dilution (dilution rate: 0.0313; Fig. 4B, arrow) or less of the PTX concentration. These findings indicate that evaluation of the PTX concentration in a test specimen should be conducted for quality control by mixing at a dilution rate of 0.0313 (32-fold dilution).

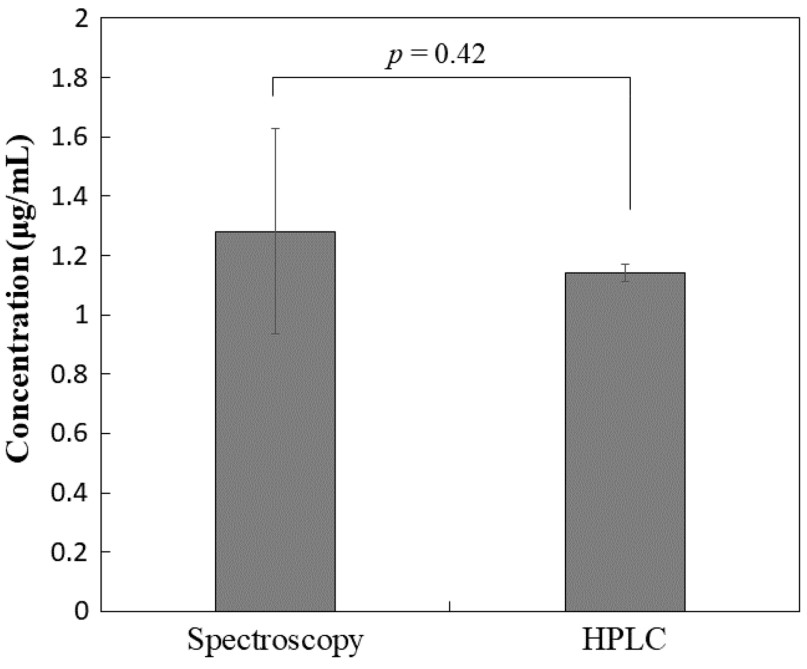

**Figure 5 Comparison of the concentration.**

## Application

According to the simulation described above, the commercially available PTX formulation can be quantitatively analyzed without the HPLC method. First, PTX after mixing (100-fold dilution with saline) should be further diluted 32-fold with saline before spectrophotometrically measuring the absorbance at a wavelength of 230 nm ($As$), which produced a reading of $0.372 \pm 0.0168$ in this study. The absorbance of the reference solution without PTX at a dilution rate of 0.0313 (32-fold dilution) should be unchanged between lots and can be measured in advance ($Ar$), producing a reading of $0.307 \pm 0.00814$ in this study. The difference between $As$ and $Ar$ ($As–Ar$: $0.0654 \pm 0.0168$) would provide the absorbance of PTX in the test specimen at a dilution rate of 0.0313, and the PTX concentration ($1.28 \pm 0.346$ µg/mL) can subsequently be determined using a calibration curve prepared based on dilution in 65.8% methanol. Comparison with the results obtained using HPLC showed that there was no significant difference between values (Fig. 5). Figure 6 shows a schematic illustration of the methodology established in this study.

## DISCUSSION

Paclitaxel can be more accurately quantified using a calibration curve when prepared in a methanol/water mixture without the need for special devices or techniques. Solvent interactions have been extensively studied since the 1970s and are known to affect not only the solubility but also the stability and reaction rate of a solute (*Hynes, 1985*; *Reichardt, 1982*). Therefore, it is important to evaluate the interactions of hydrophobic drugs such as PTX in polar aqueous solvents. In most cases, calibration standards for hydrophobic

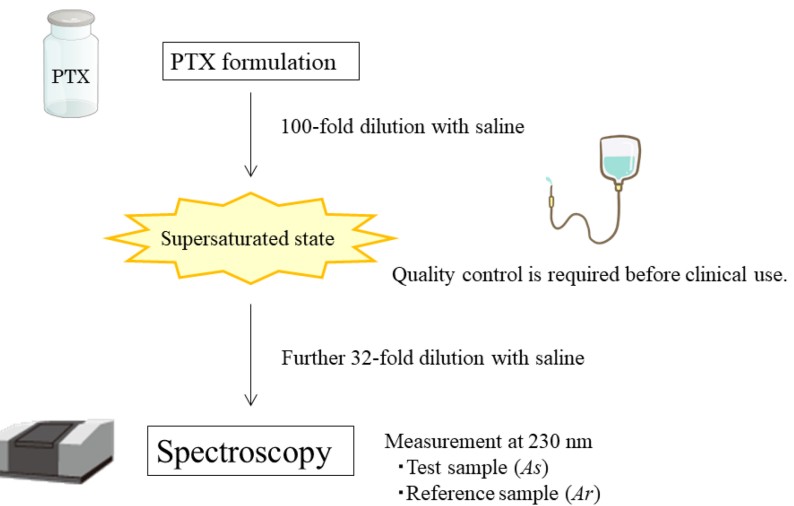

Using $(As-Ar)$ values, equation (y = 0.0486x + 0.0032) is solved.

**Figure 6 Schematic illustration of the methodology (Graphics credit: Kenkyu.net and Illustrain).**

drugs are prepared in non-polar solvents, aprotic polar solvents, and certain alcohols. However, these standards may not be accurate when evaluated using a spectrophotometer due to changes in the absorbance spectra as a result of fundamental solute/solvent interactions or other factors. For verification, methanol and acetonitrile with higher permeability in the target wavelength region (230 nm in this study) may be more suitable than DMSO and DMF. In particular, methanol is less expensive and more friendly to the environment than acetonitrile. Figure 2 shows that a marked change in the calibration curve was particularly observed at 70–80% methanol. Wakisaka and Ohki showed that the hydrogen bonding network (cluster level) in alcohol/water mixtures changes depending on the alcohol concentration, and causes various solvent effects (*Wakisaka & Ohki, 2005*). They found marked cluster-level changes at alcohol concentrations of 5.00–52.3% and 79.3–100%. As the alcohol concentration increased, clusters of water molecules formed in the former range, while clusters of alcohol molecules formed in the 52.3–79.3% range, and then disappeared in the latter range. Because a lipophilic solute is more stable when surrounded by clusters of alcohol molecules, PTX is expected to be more stable at methanol concentrations between 52.3% and 79.3%. We therefore speculate that the absorbance value increased at higher methanol concentrations due to instability, leading to marked changes in the calibration curve. Our findings suggest that quantification results for PTX, particularly at low concentrations, may vary substantially if experiments are conducted in solvents that differ to those used for calibration.

As shown in Fig. 3, we evaluated the relative error from HPLC results, and 65.8% was identified as the optimum methanol concentration for quantification of PTX. The calibration curve prepared using 0.313, 0.625, 1.25 and 2.50 μg/mL PTX in 65.8% methanol as the solvent can therefore be expressed using the regression curve with the slope 0.0486 and the intercept 0.0032.

On the other hand, PTX formulations contain polyoxyethylene castor oil and anhydrous ethanol, and the effects of these solvents should be considered when quantifying PTX. As shown in Fig. 4, there was higher correlation between the results obtained using the calibration curve and HPLC at a 32-fold dilution or less of the PTX concentration. These findings indicated that it was possible to predict the PTX concentration even when polyoxyethylene castor oil and anhydrous ethanol were added, as in the commercially available PTX formulation, by diluting 32-fold with saline after mixing, and the accuracy of the methods established in this study (Fig. 6) was verified using the commercially available PTX formulation (Fig. 5).

Although the results may differ depending on the chain length of the polyoxyethylene castor oil, the theory and process should be similar for measuring other PTX solutions. While the Beer-Lambert law supports use of the additivity of absorbance for each component in the mixture, in practice, it is necessary to verify whether the drugs and other components follow the law of additivity of absorbance, including the presence or absence of interactions, even if they are commercially available in mixed form, such as PTX formulations. To overcome the need to verify this in our present study, we established an effective method for quantifying PTX in the supersaturated state in saline based on correlations with the HPLC results, and determined the required conditions for measurement using a calibration curve.

## CONCLUSIONS

We evaluated a simple and rapid method for determining the concentration of PTX in aqueous solvent using spectrophotometry. Use of a calibration curve prepared based on dilution in 65.8% methanol was effective for analyzing the PTX concentration in saline while minimizing the solvent effect. Even when polyoxyethylene castor oil and anhydrous ethanol were added, as in the commercially available PTX formulation, it was possible to predict the PTX concentration by diluting 32-fold after mixing. This approach may be useful for quality control of PTX before clinical use.

## ACKNOWLEDGEMENTS

We would like to express our gratitude to Dr. Chris Arakawa from the University of Washington for his technical advice.

### Funding

This work was supported by JSPS KAKENHI (Grant Numbers 264086 and 26750152). The funders had no role in study design, data collection and analysis, decision to publish, or preparation of the manuscript.

### Grant Disclosures

The following grant information was disclosed by the authors:
JSPS KAKENHI: 264086 and 26750152.

## Competing Interests

The authors declare that they have no competing interests.

## Author Contributions

- Ken Sugo conceived and designed the experiments, performed the experiments, analyzed the data, performed the computation work, prepared figures and/or tables, authored or reviewed drafts of the paper, and approved the final draft.
- Mitsuhiro Ebara conceived and designed the experiments, prepared figures and/or tables, authored or reviewed drafts of the paper, and approved the final draft.

## Data Availability

The raw measurements are available in a Supplemental File.

## Supplemental Information

Supplemental information for this article can be found online at http://dx.doi.org/10.7717/peerj-achem.3#supplemental-information.

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
