# Peer review of "A simple spectrophotometric evaluation method for the hydrophobic anticancer drug paclitaxel"

_PeerJ Analytical Chemistry, doi:10.7717/peerj-achem.3_

## Round 0.1 · original submission · Major Revisions

· Academic Editor

Major Revisions

Please revise the manuscript according to the reviewers' comments. Any revisions should be clearly highlighted so that changes are easily visible to the editors and reviewers. Please include in your rebuttal if you found it impossible or inappropriate to address certain comments.

Reviewer 1 ·

Basic reporting

I find a report about the spectrophotometric determination of PTX (International Journal of Advances in Pharmaceutical Sciences 2(1):29-32, 2011). please explain about the advantage of your work than this report.

Experimental design

It is good.

Validity of the findings

Some analytical parameters of the proposed method such as limit of detection and quantification are not calculated.

Additional comments

The manuscript well designed. The below references about the spectrophotomeric and HPLC determinations can be added to the introduction.
1) Heydari, R., Hosseini, M., Alimoradi, M., Zarabi, S. A simple method for simultaneous spectrophotometric determination of brilliant blue FCF and sunset yellow FCF in food samples after cloud point extraction. Journal of the Chemical Society of Pakistan, 38, 2016, 438-445
2) Heydari, R., Hosseini, M., Zarabi, S. A simple method for determination of carmine in food samples based on cloud point extraction and spectrophotometric detection. Spectrochimica Acta - Part A: Molecular and Biomolecular Spectroscopy, 150, 2015, 786-791
3) R. Heydaria, F. Bastami, M. Hosseini, M. Alimoradi, Simultaneous determination of Tropaeolin O and brilliant blue in food samples after cloud point extraction. Iranian Chemical Communication, 5, 2017, 242-251
4) Heydari, R., Shamsipur, M., Naleini, N. Simultaneous determination of EDTA, sorbic acid, and diclofenac sodium in pharmaceutical preparations using high-performance liquid chromatography. AAPS PharmSciTech, 14, 2013, 764-769

Reviewer 2 ·

Basic reporting

In some parts of the article English must be improve and some sentences should be rewritten to be more understandable.
I consider some more references must be added.

Experimental design

Some comments and suggestions:

Introduction
In my opinion, the introduction should be more complete. For example, it can be completed taking into account these considerations:
- Authors can cite some more new references. In line 56 you can cite some more current article about the determination of paclitaxel by HPLC.
- I your article you demonstrate that PTX can be quantified spectrophotometrically. However this method will be only useful when one analyte is present in the sample, for example in drugs For the determination of PTX in plasma you will need HPLC. In my opinion in the introduction you should clarify this point.
- There are other articles where spectrophotometric methods have been proposed to quantify PTX. You should mention them in the introduction and underline the advantages of your method over these.
Figure 1 has a lot of test to be a Figure. The test should be remove from the Figure and placed in the test.

Materials and methods
- In line 114 you mention, “The concentration of PTX in each test sample was subsequently determined using the same conditions as those used to generate the standard calibration curve”, however you have not mentioned before what the test samples are. I think you should explain before what you will use as test samples.
- Line 156 and 157 describe some of the results found: “there was no significant difference between the spectrometry and HPLC values”, in my opinion these comment must be placed in Results.

Results
- In Figure 2 the title of the X Axis should be PTX concentration (µg/mL) and clarify that 40 % , 50 %, 60 % is the % of methanol. Moreover, you can use different colours for a better differentiation between the calibrations curves. Moreover, it would be useful if you add the slope and intercept value in this figure or at least in the table related to this figure.

- In “Calibration curve” you should clarify if you have used methanol and the different % of methanol (in each case) as blank or not. Because the absorbance of the different dissolvent should be taken into account. This analyte can have different absorbance in the different solvents but also the solvents themselves may have different absorbance. I consider this point should be clarify in the article.
. I also miss a figure with the spectrum of PTX in different solvents. Could it be added in a new figure?
- In line 177 and 178 as it is written it is not clear if you have used several samples of PTX-saturated saline solution with different % of methanol, if you have used the different calibration curves prepared in these solvents, or both things to be in the same solvent the sample and the calibration curve. I think this sentence should be rewritten.
- In Figure 3, the title of the X Axis should be MeOH concentration (%)
- Line 193: the regression curve should be express with the slope and intercept deviation.
- Figure 4A: PTX (+) or PTX (-) must be replace by dilution with or without PTX in order to be more understandable.

Finally, I consider that if you are proposing a new analytical method, you should provide some analytical parameters such as the detection and quantification limit.

Validity of the findings

Authors must clarify the novelty of this methods and compare it with other spectrophotometric methods previously published.

Additional comments

This paper describe a simple study with basic experiments for the determination of paclitaxel. Some important parameters have been appropriately optimized and the study was well lead. However, some important issues must be clarify and improve in order to make the article more completed and understandable.

Annotated reviews are not available for download in order to protect the identity of reviewers who chose to remain anonymous.

Reviewer 3 ·

Basic reporting

no comment

Experimental design

no comment

Validity of the findings

no comment

Additional comments

The article is important and reports interesting results for accurately quantification of hydrophobic anticancer paclitaxel (PTX) concentrations. No scientific changes required.
Some minor recommendations:
1. Line 56-57: Add more recent references to methods for quantification od PTX (maybe something from 2010-2019)
2. The sensitivity of the method depends on the amount of organic solvent (Fig. 2) please add to the text in the chapter „Calibration curve” the values of the sensitivity factor
3. Table 1: Add quantitative concentration from spectrophotometry measurement

---

## Round 0.2 · accepted · Accept

· Academic Editor

Accept

Please consider last comments from referee 1 during proof edition, thanks

Reviewer 2 ·

Basic reporting

ok

Experimental design

ok

Validity of the findings

ok

Additional comments

I consider this article can be accepted only considering these minor aspects.
- You have added the value of the limit of detection and quantification, however you should add which criterium you have used to calculate it (IUPAC, Clayton, Long and Winerfordner, etc)

- On the other hand, regression curve continue without be expressed with the corresponding deviation